# Learning Forward Compatible Representation in Class Incremental Learning by Increasing Effective Rank

## Abstract

Class Incremental Learning (CIL) is a prominent subfield of continual learning, aiming to enable models to incrementally learn new tasks while preserving the knowledge learned from the previous tasks. The main challenge of CIL is known as catastrophic forgetting, where a model that is naively fine-tuned to new tasks experiences a significant drop in performance on previous tasks. To address the challenge, previous studies have mostly focused on backward compatible approaches. Recently, a forward compatible approach has been introduced that supports a concurrent use with the existing backward compatible methods. The forward compatible method, however, is limited in that it relies solely on class information. In this study, we propose an effective-Rank based Forward Compatible (RFC) representation regularization that is not confined to specific types of information, such as class information. The proposed method increases the efficient rank of representation during the base session, thereby facilitating the encoding of more informative features pertinent to unseen novel tasks. To substantiate the effectiveness of our method, we establish a theoretical connection between effective rank and Shannon entropy of the representations. Subsequently, we conduct comprehensive experiments, by integrating it into ten well-known backward compatible CIL methods. The results demonstrate that our forward compatible approach is effective in enhancing the performance of novel tasks while mitigating catastrophic forgetting. Furthermore, the results indicate that our method significantly improves the average incremental accuracy of all ten cases that we have examined, underscoring its efficacy and general applicability.

## 1 Introduction

Continual learning, the process of continually acquiring and integrating new knowledge, has emerged as a significant challenge in the field of machine learning. In contrast to conventional learning paradigms that focus on static datasets and fixed tasks, continual learning encompasses the dynamic and ever-changing nature of real-world applications. To address this challenge, class incremental learning (CIL) (Rebuffi et al., 2017; Hou et al., 2019; Douillard et al., 2020; Shi et al., 2022), a subfield of continual learning, primarily focuses on developing techniques that enable adaptive learning to accommodate new classes, while minimizing the detrimental impact of knowledge degradation on previously learned classes.

In CIL, training takes place through multiple incremental sessions, each focusing on a distinct subset of classes that do not overlap with the subsets utilized in other sessions. The *base session* involves training a model from scratch to perform a *base task*, typically involving a large number of classes. Subsequently, in each *novel session*, the model is expected to incrementally learn a new *novel task*, typically involving a smaller number of classes, while retaining its performance on previously learned tasks. The model's performance is evaluated using the classes of all the previous and current tasks, without access to the task identification information.

The primary objective in CIL is to mitigate the detrimental impact of catastrophic forgetting, which refers to a significant decline in the model's performance on previous task classes after learning new classes from a subsequent task. To address this challenge, most of the previous works have focused on addressing the forgetting problem in the updated model (i.e., backward compatible ap-

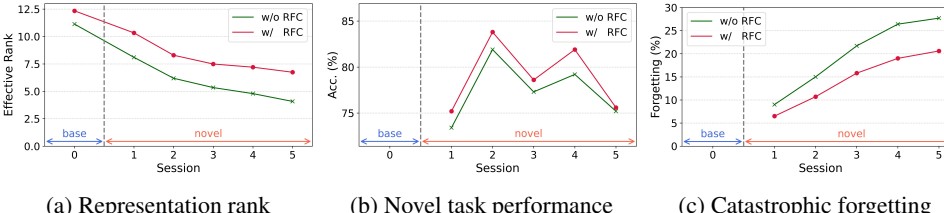

(a) Representation rank     (b) Novel task performance     (c) Catastrophic forgetting

Figure 1: Impact of increasing representation rank during the base session. We conduct an analysis of UCIR models with and without the integration of our method. ResNet-18 model is trained for the CIFAR-100 dataset, utilizing 50 base classes and a split size of 10 classes for each novel session. (a) Effective rank of the feature extractor. (b) Novel task performance in each novel session. (c) The degree of catastrophic forgetting that occurs for the base task.

proach) (Zhou et al., 2022; Shi et al., 2022). This is commonly achieved by enforcing similarity between the updated model and its predecessor. For instance, weight regularization methods enforce similarity in weights (Kirkpatrick et al., 2017; Zenke et al., 2017; Aljundi et al., 2018; Chaudhry et al., 2018), while knowledge distillation methods enforce similarity in representations (Rebuffi et al., 2017; Li & Hoiem, 2017; Castro et al., 2018; Hou et al., 2019; Wu et al., 2019).

In contrast, the concept of forward compatible approach, which aims to facilitate the training of the subsequent tasks, has received relatively little attention. While a recent work has proposed a method that learns forward compatible representations by enforcing class-wise decorrelations (CwD) (Shi et al., 2022), it has an inherent limitation as it relies on class information. Given that the training data for the base task can encompass considerably more informative features that can be utilized for novel sessions beyond mere class information, the regularization of the representation based on only the class information of the base task can impose a substantial limitation on the encoded features. In few-shot class incremental learning (FSCIL), another method called FACT also implemented forward compatible method by assigning virtual prototypes that also rely on class information (Zhou et al., 2022).

To learn forward compatible representation, this study focuses on representation rank. To be precise, we utilize *effective rank* (Roy & Vetterli, 2007) in lieu of algebraic rank. Effective rank is a continuous-value extension of algebraic rank. In contrast to algebraic rank, it possesses two advantageous properties: differentiability and the ability to effectively manage extremely small singular values. Representation rank is a general property of representations, and it is not constrained to specific types of information such as class information. Therefore, we conjecture that the representation rank can serve as a crucial indicator of the quantity of encoded features in the representation, with higher-rank representations expected to contain richer features that can be beneficial for subsequent tasks. We substantiate the conjecture with a theorem and empirical investigations. We prove that the Shannon entropy of the representation is maximized when the effective rank is maximized. Because entropy is a quantitative measure of information, larger entropy can be interpreted as richer features. Empirically, we show that effective rank increases as more classes are included in a plain supervised learning and we also show that effective rank increases as unsupervised learning proceeds.

To this end, we propose an effective-Rank based Forward Compatible (RFC) representation regularization method that increases the effective rank of representation during the base session in order to preserve informative features. Specifically, we highlight the importance of regularizing the feature extractor during the base session, as it offers two main advantages in the process of learning novel sessions. First, the performance of novel tasks can be enhanced by utilizing the rich features encoded by the base task. Second, catastrophic forgetting can be mitigated because novel tasks can leverage the rich features encoded by the base task, leading to minimal modifications to the feature extractor during the learning of novel tasks.

We have performed extensive experiments to confirm the effectiveness of RFC for class incremental learning. While we defer the explanation of the full results until Section 4, a glimpse of the experimental results is provided in Figure 1. Our method can effectively increase the representation rank as shown in Figure 1(a), can improve the performance of novel tasks as exhibited in Figure 1(b), and can substantially mitigate catastrophic forgetting as demonstrated in Figure 1(c). These empirical results provide compelling evidence for the effectiveness of our approach in learning forward compatible representation.

## 2 RELATED WORKS

The weight regularization methods aim to minimize the weight distance between the feature extractor learned in the previous session and the feature extractor learned in the subsequent session. In this approach, previous works have primarily focused on calculating the importance of each weight to penalize changes of individual weights. To calculate the importance, several methods have been proposed. EWC (Kirkpatrick et al., 2017) proposed a diagonal approximation of the Fisher Information Matrix. SI (Zenke et al., 2017) and MAS (Aljundi et al., 2018) proposed a path integral approach, which accumulates the changes in weights throughout the entire learning trajectory. RWalk (Chaudhry et al., 2018) combined the Fisher Information Matrix approach with the path integral approach.

The representation regularization methods aim to prevent forgetting by imposing a penalty on changes in representations. Typically, a regularization is applied during novel sessions, wherein Knowledge Distillation (Hinton et al., 2015) plays a key role. The previous session's network acts as the teacher, imparting its knowledge to the student network being trained in the novel session. One notable approach, iCaRL (Rebuffi et al., 2017), employs sigmoid output for knowledge distillation, while other methods (Li & Hoiem, 2017; Castro et al., 2018; Wu et al., 2019; Zhao et al., 2020) utilize temperature-scaled softmax outputs. UCIR (Hou et al., 2019) adopts cosine normalized features in knowledge distillation to alleviate biases towards new classes. PODNet (Douillard et al., 2020) effectively reduces the difference in pooled intermediate features along the height and width directions through knowledge distillation.

In the pursuit of establishing forward compatibility within class incremental learning scenarios, the Classwise Decorrelation (CwD) (Shi et al., 2022) method was developed to mimic the behavior of an oracle during the base session. Through empirical investigations, it was discerned that resembling the representation distribution patterns akin to those exhibited by the oracle - characterized by a uniform dispersion of eigenvalues across each class - holds the potential to enhance forward compatibility. To realize this, classwise Frobenius norm of representations was strategically employed during the base session, serving as a mechanism to enforce the desired distribution consistency. A known implementational limitation of CwD is that it can be challenging to employ CwD when the number of classes in the base task is large. Because of its class-wise operation, it requires a large batch-size to perform classwise decorrelation that requires reliable estimation on a per-class basis. CwD is the only forward compatible approach for CIL with our best knowledge.

If we consider a broader research area of continual learning, ForwArd Compatible Training (FACT) (Zhou et al., 2022) also introduced a forward compatible approach for few-shot class incremental learning. Within the FACT framework, the concept of virtual classes was incorporated during the training of the base session, effectively allocating embedding space to accommodate upcoming classes. FACT integrated pseudo labels and virtual instances to facilitate effective network training generated through the manifold mixup technique. These elements collectively enhanced the network's adaptability, underscoring FACT's significance in fostering the seamless integration of new classes while upholding established knowledge. FACT also relies on class information.

## 3 ENHANCING FEATURE RICHNESS BY INCREASING REPRESENTATION RANK

In this section, we present comprehensive details of the proposed method, elucidating its theoretical underpinning and substantiating its empirical implications in enhancing the feature richness of the feature extractor. The primary objective of our method is to encourage a forward compatible representation by increasing the rank of the feature extractor's output representation during the base session.

### 3.1 EFFECTIVE RANK

For a set of $N$ samples in a mini-batch, each having a $L_2$-normalized representation vector $\boldsymbol{h}_i \in \mathbb{R}^d$ satisfying $||\boldsymbol{h}_i||_2 = 1$ and $N > d$, the rank of representation matrix $\boldsymbol{H} = [\boldsymbol{h}_1, \boldsymbol{h}_2, ..., \boldsymbol{h}_N]^T \in \mathbb{R}^{N \times d}$ can be quantified as

$$\text{rank}(\boldsymbol{H}) = \text{rank}(\boldsymbol{U}\boldsymbol{\Sigma}\boldsymbol{V}^T) = \text{rank}(\boldsymbol{\Sigma}) = \sum_{i=1}^{d} \mathbf{1}_{0 < \sigma_i}, \tag{1}$$

where $\boldsymbol{U\Sigma V}^T$ is a singular value decomposition of $\boldsymbol{H}$ and $\{\sigma_i\}$ are the singular values arranged in a descending order.

The definition of algebraic rank in Eq. (1) exhibits two practical problems. The first problem is that it equally counts all positive singular values regardless of their strength. Therefore, it can be misleading when extremely small $\sigma_i$ values exist. For instance, rank is known to be susceptible to noise (Choi et al., 2017). A commonly adopted remedy for this problem is to set a threshold for counting. We call this *thresholded rank* as trank, and it is defined as

$$\mathsf{trank}(\boldsymbol{H}, \rho) \triangleq \underset{k}{\arg\min} \left( \rho \cdot \sum_{i=1}^{d} \sigma_i^2 \leq \sum_{i=1}^{k} \sigma_i^2 \right), \tag{2}$$

where $\rho$ is a threshold parameter chosen between 0 and 1. Eq. (2) quantifies the count of the largest singular values that collectively encompass $\rho$ proportion of the total singular value energy. The trank provides a straightforward way to avoid the first practical problem, but it is still vulnerable to the second problem. The second problem is the non-differentiable nature of algebraic rank and trank. They are both integer-valued and thus not differentiable. A direct consequence is the difficulty for integrating rank or trank into the end-to-end learning. An elegant work-around for this problem is known as *effective rank* (Roy & Vetterli, 2007). Effective rank is defined as

$$\mathsf{erank}(\boldsymbol{H}) \triangleq \exp\left( -\sum_{i=1}^{d} \lambda_i \log \lambda_i \right), \tag{3}$$

where $\lambda_i \triangleq \sigma_i^2/N$ and $\{\lambda_i\}$ corresponds to the set of eigenvalues for $\boldsymbol{H}^T\boldsymbol{H}/N$. We note that $\sum_{i=1}^{d} \lambda_i = 1$ because $\sum_{i=1}^{d} \lambda_i = \sum_{i=1}^{d} \sigma_i^2/N = tr(\boldsymbol{H}^T\boldsymbol{H}/N) = tr(\sum_{i=1}^{N} \boldsymbol{h}_i\boldsymbol{h}_i^T/N) = \sum_{i=1}^{N} tr(\boldsymbol{h}_i\boldsymbol{h}_i^T)/N = 1$. While being continuous, the effective rank is known to satisfy a list of properties (Roy & Vetterli, 2007). In particular, the following lower bound can be derived.

$$\mathsf{erank}(\boldsymbol{H}) \leq \mathsf{rank}(\boldsymbol{H}). \tag{4}$$

The logarithm of erank turns out to be the same as the von Neumann entropy (Nielsen & Chuang, 2002; Wilde, 2013) and its use for learning representation has been extensively studied in (Kim et al., 2023). In our work, we adopt erank as the main method for controlling feature richness.

### 3.2 PROPOSED METHOD

In class incremental learning, the training process is divided into the initial base session and the following novel sessions. Our goal is to enhance feature richness during the base session by increasing representation rank. We adopt erank as the starting point because it is differentiable. Then, we make an adjustment where we apply a logarithm because of the implementational effectiveness demonstrated in (Kim et al., 2023). The effective-Rank based Forward Compatible (RFC) representation regularization is implemented by including the RFC loss, $\mathcal{L}_{RFC}$, during the base session as below.

$$\mathcal{L} = \mathcal{L}_{CrossEntropy} + \mathcal{L}_{RFC} = \mathcal{L}_{CrossEntropy} + \alpha \cdot \sum_{i=1}^{d} \lambda_i \log \lambda_i, \tag{5}$$

where $\alpha > 0$ is the strength hyper-parameter. The RFC loss is not applied in the novel sessions, and this decision is analyzed in Section 5.2.

### 3.3 MAXIMIZATION OF SHANNON ENTROPY

Shannon entropy is a fundamental measure of information and it quantifies the information contained in a random variable (Cover, 1999). Therefore, entropy of representation can serve as a measure of feature richness. Assuming Gaussian distribution, we prove a theoretical connection between the proposed RFC and Shannon entropy.

**Theorem 1.** *For representation $\boldsymbol{h} \in \mathbb{R}^d$ that follows a multivariate Gaussian distribution, the entropy of representation is maximized if the effective rank of the representation is maximized.*

Refer to Supplementary A for the proof. It is noteworthy that the Gaussian assumption on the representation (Kingma & Welling, 2013; Yang et al., 2021) has not only been empirically observed by numerous researchers but has also led to its theoretical justifications, including (Williams, 1997; Neal, 2012; Lee et al., 2017; Yang, 2019).

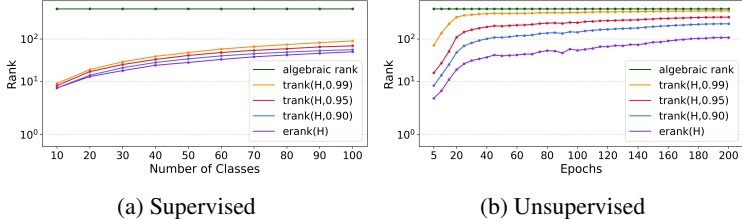

|                     |                     |
| :-----------------: | :-----------------: |
| (a) Supervised      | (b) Unsupervised    |

Figure 2: Rank vs. feature richness. ResNet-18 was trained using ImageNet-100 dataset. (a) Supervised learning – as we include more classes in the training starting from scratch, both trank and erank increase. Algebraic rank remains at the maximum value. (b) Unsupervised learning with SimCLR loss – as the unsupervised representation learning proceeds, both trank and erank increase. Algebraic rank remains at the maximum value.

### 3.4 EMPIRICAL INVESTIGATION OF RANK VS. FEATURE RICHNESS

To understand the relation between the three types of rank and feature richness in representation, we have devised two experiments. For the supervised learning experiment, we have controlled the number of classes used for the training and measured the rank values. For the unsupervised learning experiment, we have performed contrastive learning with SimCLR loss (Chen et al., 2020) and measured the rank values as the learning epoch increases. The results are shown in Figure 2. In both experiments, algebraic rank is fixed at the maximum value ($d = 512$) because it counts even extremely small singular values. Both trank and erank, however, increase as more classes are used for plain supervised learning or as training epoch increases for unsupervised learning. As we will show in the next section, a strong relationship between erank and feature richness also holds for class incremental learning.

## 4 EXPERIMENTS

We provide an overview of our experimental settings in Section 4.1. Subsequently, we demonstrate the efficacy of our method for promoting forward compatibility in Section 4.2 and for mitigating catastrophic forgetting in Section 4.3. In Section 4.4, we demonstrate that our method can improve performance of ten well-known backward-compatible methods.

### 4.1 SETTINGS

**Datasets:** We employ CIFAR-100 (Krizhevsky et al., 2009) and ImageNet-100 (Russakovsky et al., 2015), two widely adopted benchmark datasets in CIL. To maintain consistency with prior studies, we follow the standard class orderings proposed in (Rebuffi et al., 2017) for all evaluations except for the evaluation of PODNet (Douillard et al., 2020), for which we utilize the class orderings defined in PODNet. To evaluate the CIL methods, all classes in datasets are divided into multiple tasks. The initial 50 classes are designated for the base task, while the remaining classes are split into novel tasks, where the size of each split is either 10, 5, or 2. In this study, we denote split sizes of 10, 5, and 2 as S=10, S=5, and S=2, respectively.

**Implementation Details:** In this study, ResNet-18 (He et al., 2016) is employed as the base model to investigate. To reproduce the results of BiC (Wu et al., 2019), EEIL (Castro et al., 2018), iCaRL (Rebuffi et al., 2017), IL2M (Belouadah & Popescu, 2019), LwF (Li & Hoiem, 2017), MAS (Aljundi et al., 2018), SI (Zenke et al., 2017), RWalk (Chaudhry et al., 2018), and UCIR (Hou et al., 2019), we utilize the open-source codebase provided by FACIL (Masana et al., 2022). For PODNet (Douillard et al., 2020), we employ its own open-source codebase. To incorporate RFC into the aforementioned methods, we exclusively modify their respective loss functions during the training of the base session, as specified by Eq. (5). Meanwhile, the remaining configurations, such as the loss functions applied during the training of novel sessions and the default hyper-parameter settings, remain unchanged. To evaluate the performance of the models, we adopt the standard metric of *average incremental accuracy* (AIC) proposed in (Rebuffi et al., 2017).

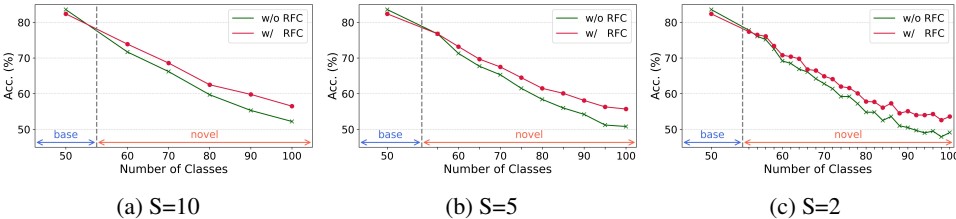

(a) S=10                  (b) S=5                  (c) S=2

Figure 3: Improvements in forward compatibility – *overall* accuracy at each session is shown for UCIR. Two ResNet-18 models are trained with and without RFC for ImageNet-100 dataset, utilizing 50 base classes under different split sizes (a) 10, (b) 5, and (c) 2 for each novel session. The feature extractors trained by the 50 classes of the base task remain **frozen** during novel sessions.

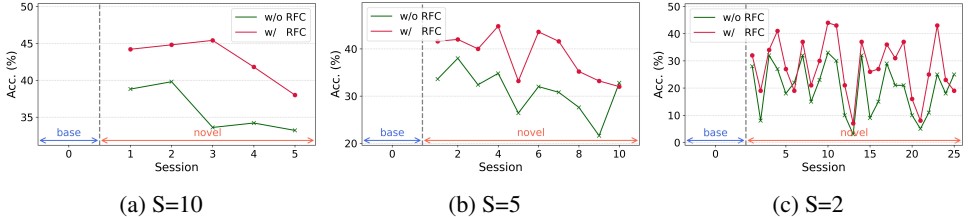

(a) S=10                  (b) S=5                  (c) S=2

Figure 4: Improvements in forward compatibility – *novel-task* accuracy at each session is shown for UCIR. Results were obtained from the same experiment as in Figure 3.

## 4.2 PROMOTING FORWARD COMPATIBILITY

To investigate the impact of our method on forward compatibility, two models are trained for a base task, one with RFC and one without RFC. Subsequently, during novel sessions, we keep the feature extractor frozen and train only the classification heads for the respective novel tasks. If increasing representation rank does indeed lead to an increase in informative features pertinent to novel tasks, it would be reasonable to expect an enhancement in performance for novel tasks. Consequently, this would serve as a supporting evidence of the forward compatible representations. The results in Figure 3 demonstrate that our method yields improved performance across all three cases: S=10 (64.78%→67.28%), S=5 (63.35%→65.98%), and S=2 (60.86%→63.45%). To delve deeper into this improvement, we analyze the performance of each individual novel task. As depicted in Figure 4, our method leads to enhanced performance for the majority of novel tasks. Moreover, the average performance in novel tasks shows substantial improvements: S=10 (43.67%→49.60%), S=5 (35.67%→42.78%), and S=2 (22.48%→30.25%). These results strongly indicate that our method is effective in promoting forward compatible representations. Additionally, we conduct a similar analysis, this time without the freezing of feature extractors. The results presented in Figure 9 and Figure 10 in Supplementary B exhibit similar performance enhancements attributed to our method, thereby substantiating the evidence regarding the forward compatibility of our method.

## 4.3 MITIGATING CATASTROPHIC FORGETTING

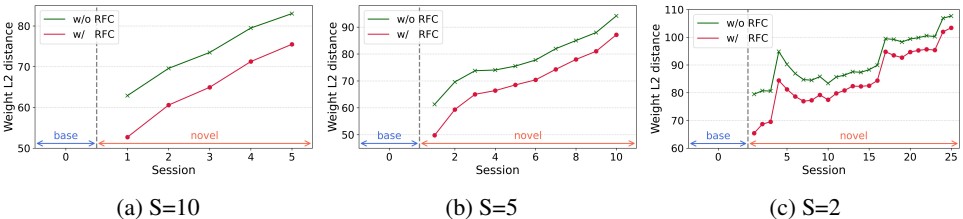

(a) S=10                  (b) S=5                  (c) S=2

Figure 5: Weight change from the base session for UCIR. Two ResNet-18 models are trained with and without RFC for ImageNet-100 dataset, utilizing 50 base classes under different split sizes (a) 10, (b) 5, and (c) 2 for each novel session.

To investigate the impact of our method on catastrophic forgetting, we conduct three comprehensive analyses. First, we examine the influence of our method on the weight changes of the feature extrac-

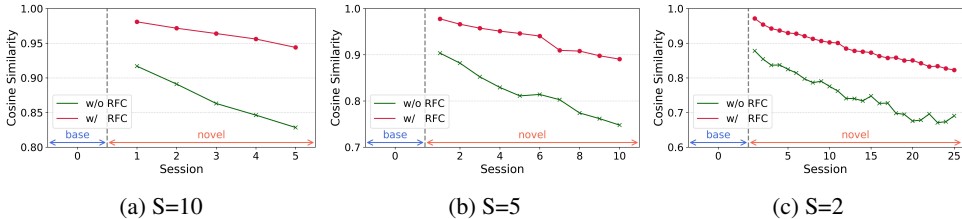

(a) S=10        (b) S=5        (c) S=2

Figure 6: Cosine similarity in representation with respect to the base session for UCIR. Two ResNet-18 models are trained with and without RFC for ImageNet-100 dataset, utilizing 50 base classes under different split sizes (a) 10, (b) 5, and (c) 2 for each novel session.

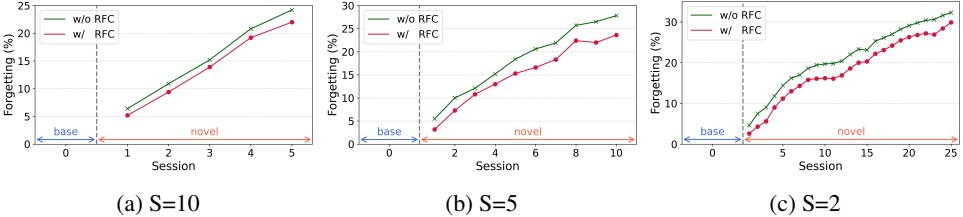

(a) S=10        (b) S=5        (c) S=2

Figure 7: Catastrophic forgetting in the base task for UCIR. Two ResNet-18 models are trained with and without RFC for ImageNet-100 dataset, utilizing 50 base classes under different split sizes (a) 10, (b) 5, and (c) 2 for each novel session.

tor. The results shown in Figure 5 indicate that when utilizing our method, the $L_2$-weight distance between the feature extractor learned in the base session and that in the novel session decreases significantly compared to the baseline. On average, there were notable reductions in weight distance over all sessions: S10: 8.70, S5: 8.13, and S2: 6.79. Additionally, Figure 11 in Supplementary B reveals that the weight distance with the feature extractor from the immediate previous session also increases less when our method is employed, showing average reduction of S10: 6.55, S5: 4.95, and S2: 2.99.

Second, we investigate the impact of our method on cosine similarity in representations produced from the validation dataset of the base task. As shown in Figure 6, our method leads to an increase in the similarity between the representations learned in the base session and those in novel sessions. The average increase in representation similarity over all sessions is significant: S10: 0.09, S5: 0.12, and S2: 0.13. Moreover, Figure 12 in Supplementary B demonstrates that the similarity with representations from the immediate previous session also increases, with average increase of S10: 0.02, S5: 0.02, and S2: 0.03.

Finally, we evaluate the actual impact on catastrophic forgetting. Figure 7 clearly illustrates that our method results in reduced catastrophic forgetting across all sessions. On average, the reductions are as follows: S10: 12.92%→11.62%, S5: 16.70%→13.86%, and S2: 20.66%→17.71%.

## 4.4 IMPROVING PERFORMANCE OF EXISTING METHODS

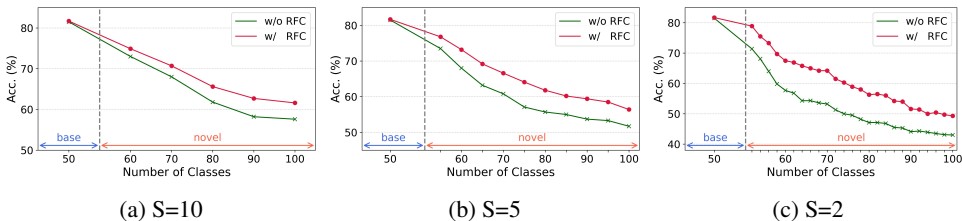

(a) S=10        (b) S=5        (c) S=2

Figure 8: Overall accuracy at each session for UCIR. Two ResNet-18 models are trained with and without RFC for *CIFAR-100 dataset*, utilizing 50 base classes under different split sizes (a) 10, (b) 5, and (c) 2 for each novel session.

To demonstrate the efficacy of RFC in enhancing performance, we conduct comprehensive experiments over ten well-known existing works. The results presented in Table 1 and Table 2 provide

Table 1: Performance improvements by RFC. All methods in this table follow the same class orderings as originally proposed in iCaRL (Rebuffi et al., 2017).

| Method | CIFAR100 (B=50) | | | ImageNet-100 (B=50) | | |
|---|---|---|---|---|---|---|
| | S=10 | S=5 | S=2 | S=10 | S=5 | S=2 |
| BiC (Wu et al., 2019) | $54.36_{\pm1.23}$ | $43.42_{\pm2.70}$ | $26.73_{\pm1.89}$ | $61.00_{\pm4.69}$ | $53.12_{\pm5.68}$ | $25.44_{\pm1.74}$ |
| with RFC | $58.06_{\pm0.73}$ | $48.02_{\pm2.33}$ | $27.74_{\pm3.11}$ | $64.18_{\pm1.94}$ | $53.77_{\pm3.05}$ | $27.56_{\pm2.72}$ |
| Improvement | +3.71 | +4.60 | +1.01 | +3.18 | +0.65 | +2.13 |
| EEIL (Castro et al., 2018) | $37.81_{\pm2.55}$ | $24.48_{\pm1.18}$ | $32.37_{\pm2.52}$ | $52.04_{\pm1.90}$ | $48.65_{\pm2.34}$ | $43.95_{\pm1.65}$ |
| with RFC | $39.52_{\pm0.38}$ | $25.79_{\pm1.18}$ | $33.46_{\pm0.69}$ | $54.74_{\pm0.91}$ | $51.64_{\pm0.80}$ | $47.89_{\pm0.60}$ |
| Improvement | +1.72 | +1.31 | +1.09 | +2.71 | +2.98 | +3.94 |
| iCaRL (Rebuffi et al., 2017) | $49.83_{\pm2.33}$ | $46.86_{\pm2.38}$ | $44.69_{\pm1.97}$ | $53.77_{\pm1.68}$ | $48.41_{\pm8.70}$ | $49.78_{\pm3.65}$ |
| with RFC | $50.56_{\pm1.47}$ | $48.48_{\pm1.17}$ | $44.99_{\pm0.97}$ | $57.17_{\pm1.66}$ | $55.62_{\pm0.71}$ | $54.40_{\pm1.63}$ |
| Improvement | +0.72 | +1.62 | +0.3 | +3.4 | +7.21 | +4.61 |
| IL2M (Belouadah & Popescu, 2019) | $40.85_{\pm2.22}$ | $24.92_{\pm1.12}$ | $33.41_{\pm2.12}$ | $56.78_{\pm1.34}$ | $52.35_{\pm2.16}$ | $47.10_{\pm1.85}$ |
| with RFC | $42.54_{\pm0.49}$ | $25.19_{\pm0.41}$ | $35.20_{\pm0.78}$ | $58.40_{\pm1.04}$ | $55.25_{\pm0.78}$ | $50.09_{\pm0.83}$ |
| Improvement | +1.69 | +0.27 | +1.78 | +1.62 | +2.9 | +2.99 |
| LwF (Li & Hoiem, 2017) | $39.25_{\pm2.04}$ | $26.53_{\pm1.87}$ | $33.41_{\pm2.06}$ | $54.26_{\pm1.32}$ | $50.30_{\pm1.96}$ | $44.52_{\pm1.70}$ |
| with RFC | $40.98_{\pm0.34}$ | $28.82_{\pm4.17}$ | $34.46_{\pm0.63}$ | $56.48_{\pm1.22}$ | $52.52_{\pm0.35}$ | $47.57_{\pm1.02}$ |
| Improvement | +1.73 | +2.29 | +1.06 | +2.22 | +2.21 | +3.05 |
| MAS (Aljundi et al., 2018) | $38.02_{\pm2.49}$ | $26.45_{\pm1.18}$ | $31.66_{\pm3.20}$ | $52.17_{\pm1.75}$ | $50.01_{\pm2.27}$ | $46.61_{\pm1.15}$ |
| with RFC | $39.54_{\pm0.71}$ | $28.80_{\pm4.52}$ | $33.36_{\pm0.52}$ | $55.04_{\pm1.30}$ | $52.77_{\pm1.02}$ | $50.32_{\pm1.38}$ |
| Improvement | +1.53 | +2.35 | +1.71 | +2.88 | +2.76 | +3.71 |
| SI (Zenke et al., 2017) | $37.90_{\pm2.39}$ | $22.88_{\pm1.41}$ | $31.01_{\pm3.24}$ | $51.89_{\pm1.70}$ | $49.25_{\pm2.19}$ | $44.16_{\pm1.64}$ |
| with RFC | $39.41_{\pm0.60}$ | $24.02_{\pm0.38}$ | $33.48_{\pm0.80}$ | $54.99_{\pm1.08}$ | $51.98_{\pm0.89}$ | $47.46_{\pm1.09}$ |
| Improvement | +1.51 | +1.13 | +2.47 | +3.1 | +2.73 | +3.29 |
| RWalk (Chaudhry et al., 2018) | $35.75_{\pm1.63}$ | $23.34_{\pm2.06}$ | $27.02_{\pm5.36}$ | $41.08_{\pm1.91}$ | $21.44_{\pm5.69}$ | $20.81_{\pm2.12}$ |
| with RFC | $39.00_{\pm1.15}$ | $24.44_{\pm1.06}$ | $32.85_{\pm1.40}$ | $44.83_{\pm1.87}$ | $35.42_{\pm1.71}$ | $22.68_{\pm4.70}$ |
| Improvement | +3.25 | +1.1 | +5.82 | +3.75 | +13.98 | +1.88 |
| UCIR (Hou et al., 2019) | $66.30_{\pm0.36}$ | $60.57_{\pm0.56}$ | $52.74_{\pm0.72}$ | $70.57_{\pm0.51}$ | $67.62_{\pm0.37}$ | $63.22_{\pm0.42}$ |
| with RFC | $69.45_{\pm0.29}$ | $66.16_{\pm0.10}$ | $61.23_{\pm0.13}$ | $71.65_{\pm0.52}$ | $69.52_{\pm0.13}$ | $65.46_{\pm0.57}$ |
| Improvement | +3.14 | +5.59 | +8.49 | +1.08 | +1.90 | +2.24 |
| Average Improvement | +2.11 | +2.25 | +2.64 | +2.66 | +4.15 | +3.09 |

compelling evidence of the significant and consistent performance improvements achieved through the integration of RFC into the previous works. Specifically, we observe notable average performance improvements for the state-of-the-art works, including a 3.74% average improvement for UCIR, a 2.63% average improvement for PODNet (NME), and a 1.24% average improvement for PODNet (CNN). Particularly, a remarkable performance improvement is observed in the case of UCIR for CIFAR-100 and its further analyses are shown in Figure 8. The achieved improvements are 3.14% increase for S=10, 5.59% for S=5, and 8.49% for S=2.

Table 2: Performance improvements by RFC for PODNet. Unlike the other works, PODNet used its own class orderings for evaluation. As an effort to make comparisons as fair as possible, we followed PODNet's class orderings for this table.

| Method | CIFAR100 (B=50) | | | ImageNet-100 (B=50) | | |
|---|---|---|---|---|---|---|
| | S=10 | S=5 | S=2 | S=10 | S=5 | S=2 |
| PODNet(NME) (Douillard et al., 2020) | $68.47_{\pm1.27}$ | $67.09_{\pm1.19}$ | $65.16_{\pm1.02}$ | $60.42_{\pm0.53}$ | $50.93_{\pm0.73}$ | $36.36_{\pm0.30}$ |
| with RFC | $69.51_{\pm0.56}$ | $67.66_{\pm0.63}$ | $65.59_{\pm0.48}$ | $64.08_{\pm0.45}$ | $56.34_{\pm0.75}$ | $41.03_{\pm2.15}$ |
| Improvement | +1.04 | +0.57 | +0.43 | +3.66 | +5.42 | +4.67 |
| PODNet(CNN) | $66.11_{\pm0.63}$ | $63.49_{\pm0.45}$ | $59.61_{\pm0.53}$ | $73.76_{\pm0.17}$ | $68.32_{\pm0.27}$ | $61.82_{\pm0.68}$ |
| with RFC | $67.13_{\pm0.86}$ | $64.72_{\pm1.01}$ | $61.49_{\pm0.71}$ | $74.42_{\pm0.27}$ | $69.66_{\pm0.09}$ | $63.11_{\pm0.22}$ |
| Improvement | +1.02 | +1.23 | +1.89 | +0.66 | +1.34 | +1.29 |
| Average Improvement | +1.03 | +0.90 | +1.16 | +2.16 | +3.38 | +2.98 |

## 5 DISCUSSION

### 5.1 COMPARISON WITH CWD

We conduct an evaluation of the efficacy of RFC in comparison to CwD (Shi et al., 2022), a forward compatible method that leverages class information. The performance evaluation is carried out using the UCIR and PODNet models with CIFAR-100 dataset. As shown in Table 3, RFC consistently

demonstrates superior performance, exhibiting considerable performance improvements of 1.76% for S=10, 2.27% for S=5, and 3.10% for S=2 on average. The improvement tends to be larger when dealing with smaller split sizes, or equivalently, when handling a larger number of novel sessions.

Table 3: Comparison with CwD.

| | CIFAR100 (B=50) | | | | | |
|---|---|---|---|---|---|---|
| | UCIR | | | PODNet (CNN) | | |
| | S=10 | S=5 | S=2 | S=10 | S=5 | S=2 |
| Baseline | $66.30_{\pm0.36}$ | $60.57_{\pm0.56}$ | $52.74_{\pm0.72}$ | $66.11_{\pm0.63}$ | $63.49_{\pm0.45}$ | $59.61_{\pm0.53}$ |
| with CwD | $67.06_{\pm0.12}$ | $62.71_{\pm0.36}$ | $56.39_{\pm0.30}$ | $66.01_{\pm0.75}$ | $63.63_{\pm1.02}$ | $60.14_{\pm1.16}$ |
| with RFC | $69.45_{\pm0.29}$ | $66.16_{\pm0.10}$ | $61.23_{\pm0.13}$ | $67.13_{\pm0.86}$ | $64.72_{\pm1.01}$ | $61.49_{\pm0.71}$ |
| Improvement | +2.39 | +3.45 | +4.84 | +1.12 | +1.09 | +1.35 |

## 5.2 INCREASING REPRESENTATION RANK DURING NOVEL SESSIONS

RFC increases effective rank during the base session only. It is also possible to increase effective rank during novel sessions with the goal of acquiring additional features that may be useful in the subsequent novel sessions. Such a strategy, however, can also intensify the modifications of the learned model from the previous sessions. To investigate the overall effect, we have conducted an experiment and the results are shown in Table 4. In short, no significant improvement was observed by increasing effective rank during novel sessions. Therefore, we have chosen to apply effective rank regularization only during the base session.

Table 4: Influence of increasing representation rank during novel sessions.

| | CIFAR100 (B=50) | | | | | | | | |
|---|---|---|---|---|---|---|---|---|---|
| | S=10 | | | S=5 | | | S=2 | | |
| | Base Only | Base+Novel | Diff. | Base Only | Base+Novel | Diff. | Base Only | Base+Novel | Diff. |
| BiC | $58.06_{\pm0.73}$ | $58.29_{\pm0.38}$ | +0.23 | $48.02_{\pm2.33}$ | $46.60_{\pm1.91}$ | -1.41 | $27.74_{\pm3.11}$ | $27.64_{\pm1.83}$ | -0.10 |
| EEIL | $39.52_{\pm0.38}$ | $39.44_{\pm0.63}$ | -0.08 | $25.79_{\pm1.18}$ | $25.62_{\pm1.10}$ | -0.17 | $33.46_{\pm0.69}$ | $33.16_{\pm1.07}$ | -0.29 |
| iCaRL | $50.56_{\pm1.47}$ | $50.50_{\pm2.04}$ | -0.06 | $48.48_{\pm1.17}$ | $48.23_{\pm1.14}$ | -0.25 | $44.99_{\pm0.97}$ | $44.81_{\pm1.18}$ | -0.18 |
| IL2M | $42.54_{\pm0.49}$ | $42.76_{\pm0.83}$ | +0.21 | $25.19_{\pm0.41}$ | $26.22_{\pm1.77}$ | +1.03 | $35.20_{\pm0.78}$ | $34.87_{\pm0.67}$ | -0.32 |
| LwF | $40.98_{\pm0.34}$ | $40.69_{\pm0.81}$ | -0.28 | $28.82_{\pm4.17}$ | $26.70_{\pm0.39}$ | -2.12 | $34.46_{\pm0.63}$ | $34.63_{\pm0.75}$ | +0.17 |
| MAS | $39.54_{\pm0.71}$ | $39.77_{\pm0.66}$ | +0.23 | $28.80_{\pm4.52}$ | $29.03_{\pm3.50}$ | +0.23 | $33.36_{\pm0.52}$ | $33.91_{\pm0.91}$ | +0.55 |
| SI | $39.41_{\pm0.60}$ | $39.51_{\pm0.58}$ | +0.10 | $24.02_{\pm0.38}$ | $23.89_{\pm0.19}$ | -0.13 | $33.48_{\pm0.80}$ | $33.60_{\pm0.52}$ | +0.12 |
| RWalk | $39.00_{\pm1.15}$ | $38.91_{\pm1.39}$ | -0.09 | $24.44_{\pm1.06}$ | $25.96_{\pm3.06}$ | +1.53 | $32.85_{\pm1.40}$ | $33.04_{\pm1.20}$ | +0.19 |
| UCIR | $69.45_{\pm0.29}$ | $69.57_{\pm0.23}$ | +0.12 | $66.16_{\pm0.10}$ | $66.55_{\pm0.16}$ | +0.39 | $61.23_{\pm0.13}$ | $61.27_{\pm0.29}$ | +0.04 |
| Average Improvement | | | +0.04 | | | -0.10 | | | +0.02 |

## 5.3 SENSITIVITY STUDY OF REGULARIZATION COEFFICIENT

We performed a sensitivity study of the strength hyper-parameter $\alpha$ in Eq. (5) on the performance. The results are presented in Figure 13 of Supplementary B, and they demonstrate consistent and smooth inverted U-shaped patterns in performance, with the peak performance observed at the value of $\alpha = 0.1$.

## 6 CONCLUSION

In this study, we propose an effective-Rank based Forward Compatible (RFC) representation regularization method that can be integrated with a wide range of existing methods in CIL. More specifically, our method increases the effective rank of representations during the base session. In order to substantiate the effectiveness of our method, we have established a theoretical connection between the effective rank and the Shannon entropy of representations. Through empirical analysis, we have demonstrated the efficacy of our method in several dimensions including enhancement in forward compatibility, mitigation of catastrophic forgetting, and improvement in performance. In summary, our method effectively enhances the performance by promoting forward compatibility of the learned representations.

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

## A   PROOF OF THEOREM

**Theorem 1.** *For representation $\boldsymbol{h} \in \mathbb{R}^d$ that follows a multivariate Gaussian distribution, the entropy of representation is maximized if the effective rank of the representation is maximized.*

*Proof.* Without loss of generality, we consider normalized representation vectors (i.e., $||\boldsymbol{h}||_2 = 1$) that follow a zero-mean multivariate Gaussian distribution (i.e., $\boldsymbol{h} \sim \mathcal{N}(0, \Sigma)$), thereby satisfying the condition $tr(\Sigma) = 1$.

The proof is based on two parts. In the first part, we prove that the solution for maximizing the representation entropy is $\Sigma = \frac{1}{d}I$. In the second part, we prove that the solution for maximizing effective rank is also $\Sigma = \frac{1}{d}I$.

The entropy of multivariate Gaussian distribution, $\mathcal{N}(\mu, \Sigma)$, can be computed as (Cover, 1999)

$$\frac{d}{2} \log 2\pi e + \frac{1}{2} \log \det(\Sigma), \tag{6}$$

where $\det(\Sigma)$ is the determinant of $\Sigma$. By examining the equation, it can be confirmed that entropy is maximized when $\det(\Sigma)$ is maximized under the constraint of $tr(\Sigma) = 1$. The solution for this optimization problem is $\frac{1}{d}I$ because of Hadamard's inequality and inequality of arithmetic and geometric means. First, the following Hadamard's inequality states that the determinant of a positive definite matrix is less than the product of its diagonal elements.

$$\det(\Sigma) \leq \prod_i \Sigma_{ii}, \text{ with equality iff } \Sigma \text{ is diagonal.} \tag{7}$$

Therefore, all the off-diagonal terms need to be zero to maximize the entropy. Second, $\lambda_i$ needs to be equal to $1/d$ for all $i$ – otherwise, the inequality of arithmetic and geometric implies that $\prod_i \Sigma_{ii}$ can be increased further while satisfying the sum constraint of $tr(\Sigma) = 1$.

The proof for $\Sigma = \frac{1}{d}I$ being the solution for maximizing the effective rank in Eq. (3) is trivial. The logarithm of effective rank is $-\sum_{i=1}^{d} \lambda_i \log \lambda_i$ where $\sum_{i=1}^{d} \lambda_i = 1$. Because this can be interpreted as the entropy of a probability distribution denoted by $\{\lambda_i\}$, it is maximized by the uniform distribution, i.e., $\lambda_i = \frac{1}{d}$. $\qquad \square$

## B   SUPPLEMENTARY RESULTS

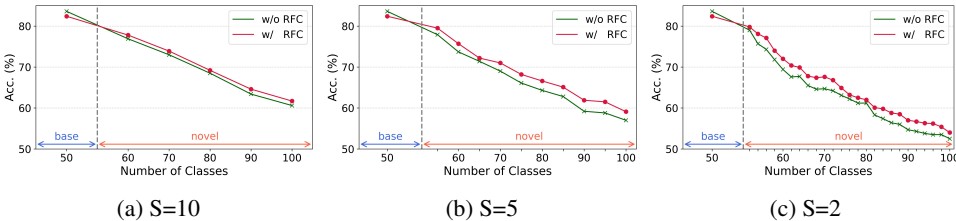

(a) S=10  (b) S=5  (c) S=2

Figure 9: Improvements in forward compatibility – *overall* accuracy at each session is shown for UCIR. Two ResNet-18 models are trained with and without RFC for ImageNet-100 dataset, utilizing 50 base classes under different split sizes (a) 10, (b) 5, and (c) 2 for each novel session.

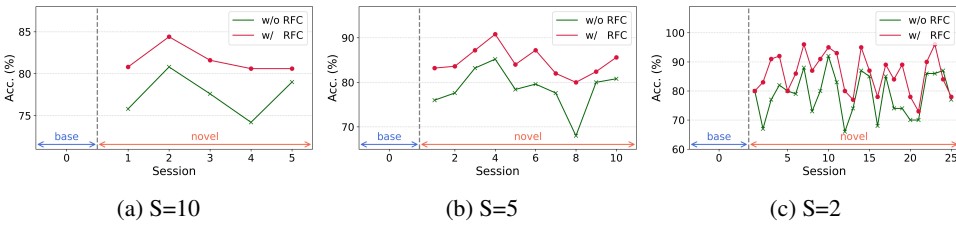

(a) S=10            (b) S=5            (c) S=2

Figure 10: Improvements in forward compatibility – *novel-task* accuracy at each session is shown for UCIR. Results were obtained from the same experiment as in Figure 9.

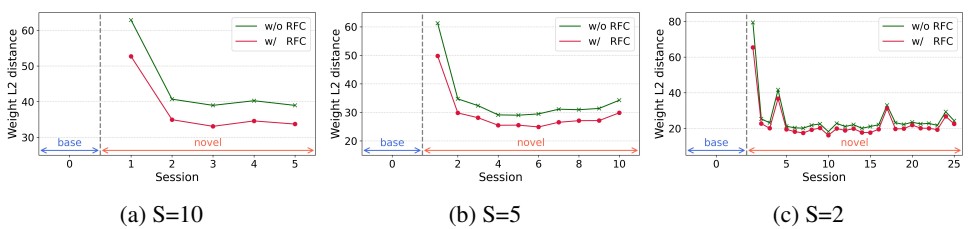

(a) S=10            (b) S=5            (c) S=2

Figure 11: Weight change from the immediate previous session for UCIR. Two ResNet-18 models are trained with and without RFC for ImageNet-100 dataset, utilizing 50 base classes under different split sizes (a) 10, (b) 5, and (c) 2 for each novel session.

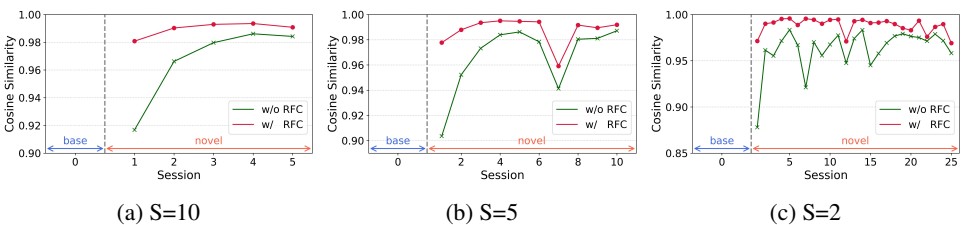

(a) S=10            (b) S=5            (c) S=2

Figure 12: Cosine similarity in representation with respect to the immediate previous session for UCIR. Two ResNet-18 models are trained with and without RFC for ImageNet-100 dataset, utilizing 50 base classes under different split sizes (a) 10, (b) 5, and (c) 2 for each novel session.

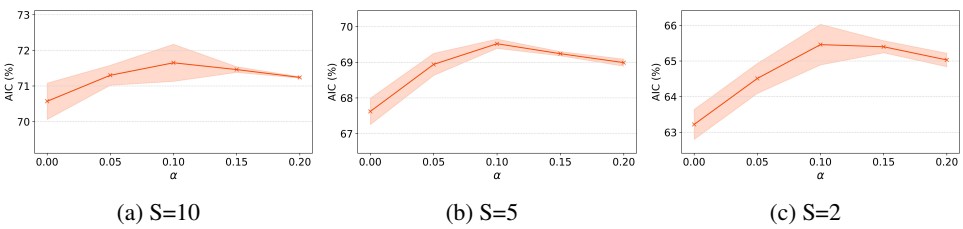

(a) S=10            (b) S=5            (c) S=2

Figure 13: Impact of regularization coefficient on the performance. ResNet-18 models are trained with a range of regularization coefficients for ImageNet-100 dataset, utilizing 50 base classes under different split sizes (a) 10, (b) 5, and (c) 2 for each novel session.

