# OpenReview forum: "Learning Forward Compatible Representation in Class Incremental Learning by Increasing Effective Rank"
_ICLR.cc/2024/Conference — ICLR 2024 Conference Withdrawn Submission_

### Official Review · Reviewer_NY2P · 2023-10-22

**Soundness:** 3 good
**Presentation:** 2 fair
**Contribution:** 2 fair
**Rating:** 3
**Confidence:** 5

**Summary:**

This paper tackles the class-incremental learning problem from the forward compatibility perspective. Class-incremental learning is a hot topic and desired capability in today’s representation learning field. The authors propose to use effective rank-based regularization terms to restrict the initial stage of training. The proposed method is evaluated on CIFAR100 and ImageNet100 against several baselines.

**Strengths:**

1. Class-incremental learning is of great importance to the machine learning field.
2. The proposed method is evaluated on CIFAR100 and ImageNet100 against several baselines.
3. The source code is attached, enhancing the reproducibility.

**Weaknesses:**

1. Overall, I find the intuition to utilize effective rank to enhance forward compatibility is not well illustrated. There are many core factors that are not well discussed in the main paper, e.g., the relationship between feature richness and effective rank, the relationship between feature richness and forward compatibility, and the relationship between feature richness/effective rank and catastrophic forgetting. Hence, reading the paper sometimes makes me confused about why the heuristic loss works for the specific capability.
2. As repeated many times in the main paper, the authors argue that CwD [1] and FACT [2] require “class information” to enhance forward compatibility. What does the “class information” mean in this paper, since the concept is abstract and hard to figure out? In my opinion, CwD and FACT can work during the first incremental stage and use only the base classes (first-stage classes) to enhance forward compatibility. They do not require extra information like future classes to design the loss term, which is almost the same as the authors do in Eq. 5. Hence, it requires careful clarification about this claim point.
3. As another noteworthy weakness, I must highlight that the current experimental evaluation is somehow weak, lacking the comparison on ImageNet1K (dataset) to other state-of-the-art works like DER, FOSTER [3-4] (methods), other related competitors like FACT, BEEF [5] (methods), and the experimental results without half base classes (settings). See more comments in the following questions.

**Questions:**

Apart from the above weaknesses, there are several other questions that need to be addressed in the rebuttal.
1. As a benchmark dataset for large-scale class-incremental learning, reporting the results on ImageNet1000 is essential for a holistic overview.
2. The baseline methods adopted in Table 1 are somehow outdated (among them, the latest is from 2019). I suggest the authors consider more recent state-of-the-art methods like DER and FOSTER [3-4] and see whether the proposed method performs well on these algorithms. Also, it would be better to show if the proposed method can work on other backbones like ViT (e.g., DyTox).
3. There are several works addressing forward compatibility in the class-incremental learning field, e.g., CwD, FACT, and BEEF [1,2,5]. However, only CwD is adopted in the comparison in Table 3. Since this paper is targeted for forward compatibility, it is essential to conduct a holistic evaluation considering all these competitors fairly.
4. Apart from training from half classes, there is another setting in class-incremental learning that equally divides all the classes into incremental stages. Under this setting, the authors cannot utilize the rich class information for regularization. It is essential to provide corresponding experimental results to show whether the proposed method can tackle various class-incremental learning settings.


In summary, this paper tackles an important problem in the class-incremental learning field, which is a desired capability in future research. However, my concerns focus on the not well-illustrated intuition, exaggerated contribution, and half-baked experiments.

[1] Mimicking the Oracle: An Initial Phase Decorrelation Approach for Class Incremental Learning. CVPR 2022

[2] Forward Compatible Few-Shot Class-Incremental Learning. CVPR 2022

[3] DER: Dynamically Expandable Representation for Class Incremental Learning. CVPR 2021

[4] Foster: Feature boosting and compression for class-incremental learning. ECCV 2022

[5] BEEF: Bi-Compatible Class-Incremental Learning via Energy-Based Expansion and Fusion. ICLR 2023

---

### Official Review · Reviewer_7g6A · 2023-10-23

**Soundness:** 2 fair
**Presentation:** 3 good
**Contribution:** 2 fair
**Rating:** 5
**Confidence:** 4

**Summary:**

The paper introduces a novel method for promoting compatible representations in the context of class incremental learning. The authors have designed a regularization term, known as RFC loss, which is based on a continuous approximation of matrix ranks referred to as the **Effective Rank**. The central claim is that maximizing the effective rank of the matrix containing feature representations enhances the feature richness. In particular, applying this regularization only to the base task within a class incremental learning process positively impacts the acquisition of new tasks, leading to improved overall performance and mitigating catastrophic forgetting. Their method falls under the category of forward-compatible approaches, but as demonstrated in the experimental section, the RFC loss can significantly enhance the performance of many common backward-compatible methods that represent the current state-of-the-art solutions for the class incremental learning problem.

**Strengths:**

1. The paper's introduction is well-written, effectively motivating and framing the problem that the paper addresses. It provides a clear context for the reader, which is crucial for understanding the proposed approach.
2. The RFC loss introduced in the paper is an elegant solution for enriching feature representation during the base task and the author provides a solid theoretical rationale to support their claims.
3. The paper provides a robust empirical foundation by conducting analyses against a baseline method. This empirical evidence demonstrates the success of the feature representation achieved through the RFC loss in improving performance across different task sequences in a class incremental learning scenario.
4. The experiments carried out in the paper consistently show that the RFC loss has a positive synergy with various state-of-the-art methods. This synergy leads to improved accuracy across different task sequences and datasets.

**Weaknesses:**

1. The RFC loss is only used during the training of the base task and does not exhibit significant improvements when applied also to novel tasks. While the authors acknowledge that using this regularization intensifies model modifications from the previous session, a more detailed analysis of this limitation is essential since it represent a crucial problem in class incremental learning.

2. Many state-of-the-art methods cited in the paper that benefit from the RFC loss are backward-compatible exemplar-based approaches. However, in recent times, exemplar-free methods have emerged as strong competitors. They rely on prototypes instead of examplars and often employ Class Augmentation techniques during the base task to enhance the initial feature representation. For example, [1] employs a mixup approach to achieve a similar effect to the proposed RFC loss.  In fact, in that paper, the authors show that their method can increase the number of directions with significant eigenvalues in the feature space. Other works, such as [2], follow similar reasoning by using self-rotation as class augmentation technique. In my opinion, a comparison with these exemplar-free methods could be valuable to understand whether RFC could be used in substitution or synergy with these Class Augmentation techniques.

3. The paper could benefit from the inclusion of a dedicated section addressing the limitations of the proposed approach. This section would provide transparency and a more thorough assessment of potential shortcomings, allowing readers to understand the practical constraints and challenges associated with the RFC loss method.

[1]: Class-Incremental Learning via Dual Augmentation, NIPS 2021

[2]: Prototype augmentation and self-supervision for incremental learning, CVPR 2021

**Questions:**

My concerns are summarized in the Weaknesses section. Here, I add two more technical questions:

1. At the beginning of Section 3.1, your method selects $N$ samples from a mini-batch, and it also required that $N > d$, where $d$ represents the dimensions of the feature space, which is 512 when ResNet18 is used as the backbone. I assume that this requirement depends on the relationship $\text{erank}(H) < \text{rank}(H) \le \min(N, d)$. However, in your provided code (```ucir_w_rfc.py```), I noticed that a batch size of 128 is employed, which seems to contradict this derivation. Could you please clarify this point? In a more general sense, does the batch size play a significant role in your approach? If so, have you conducted an ablation study on this hyperparameter? While the necessity of a large batch size may not be a primary concern, it might be worth mentioning as a potential limitation of the method if the regularization doesn't show a significant effect with smaller batch sizes.

2. Figure 2a show the differences between various rank definitions in a standard supervised learning scenario. It would be interesting to see a similar plot (or analysis) for the class-incremental scenario, both with and without the RFC regularizer. For example, at the end of the base task, depending on the presence of the regularizer, how does the effective rank compare to the theoretical rank? In my opinion, it would also be interesting to observe how the effective rank behaves when the mentioned Class Augmentation techniques are applied in comparison to when the RFC loss is applied.

For now I set my score at 5, but I'm sincerely interest in the further development of this discussion.

---

### Official Review · Reviewer_od6p · 2023-10-30

**Soundness:** 2 fair
**Presentation:** 3 good
**Contribution:** 2 fair
**Rating:** 3
**Confidence:** 4

**Summary:**

This paper proposes an approach that trains a base model by regularizing it so that it can increase the effective rank of representation to learn more informative features, which enable less forgetting.

**Strengths:**

* The proposed specific method should be new, but the idea has been around for a while and there are existing methods for the purpose.
* The writing is easy to follow.

**Weaknesses:**

*  The novelty of the paper is limited. Although the proposed specific method may be different from others, the idea is not new, and many existing methods have been proposed to achieve the same effect. Existing methods mainly achieve it by using contrastive learning and/or data augmentation, which are not compared in this paper but should be. The following paper specifically tries to increase the richness of the learned features,

[1]. Class-Incremental Learning via Dual Augmentation. NeurIPS, 2021.

* The following paper considers both forward and backward as it uses out-of-distribution detection. It also uses contrastive learning.

[2] A theoretical study on solving continual learning. NeurIPS, 2022.

* The base task in the paper is a form of pretraining. Existing method using a pre-trained model should be compared, e.g.,

[3] Lifelong machine learning with deep streaming linear discriminant analysis. CVPR workshops, 2020.
[4] Learning to prompt for continual learning. CVPR, 2022
[5] Learnability and algorithm for continual learning. ICML 2023.

* The accuracy results based on average incremental accuracy is very low, may be lower than accuracy after learning the last task of some SoTa methods.

* The baselines used in the paper are old.

* More datasets should be used in the experiments.

**Questions:**

See above

---

### Official Review · Reviewer_VnSg · 2023-11-01

**Soundness:** 3 good
**Presentation:** 3 good
**Contribution:** 2 fair
**Rating:** 5
**Confidence:** 4

**Summary:**

* This paper proposes to use a rank-based forward compatible (RFC) representation regularization to address the catastrophic forgetting in class incremental learning (CIL). The results demonstrate that this approach is effective in enhancing the performance of some existing methods.

**Strengths:**

* Increasing the rank of representation for continual learning is sound and interesting.
* The proposed method is well verified by illustrative figures and experiments results.

**Weaknesses:**

* Compared methods. The used baselines in this paper is old. iCaRL, EEIL, BiC, UCIR are early-stage exemplar-replay methods, and stronger baselines like DER are suggested to included. Similarly, LwF, MAS, SI, Rwalk are early-stage regularization methods, and stronger non-exemplar based methods are suggested to be included.
* Experiments. How the proposed method performs on large scale dataset ImageNet Full?
* Novelty. The proposed method is the same as vne proposed in (Kim et al., 2023), which may lack of novelty. In additional, the idea of increasing rank for incremental learning has been explored in the paper of CwD  (Shi et al., 2022).

**Questions:**

* In Fig.2, the rank is increasing when learning more classes. In Fig.1 (a), the effective rank is decreasing when learning more classes. Is the effective rank computed on the representations of based classes, or all learned classes?

---

### Author Response · Authors · 2023-11-16
**Common response**

Dear all reviewers,

We express our sincere gratitude for your feedback. To address your constructive comments thoroughly and enhance the quality of our work, we have made the decision to withdraw the submission at this time.

In our revised manuscript, we will incorporate more comprehensive experiments based on your feedback. This will involve a thorough comparison of RFC with methods that specifically address forward compatibility, conducted on a larger scale dataset.

By the way, we would like to clarify one thing. The primary objective of our study is to enhance forward compatibility in the base feature extractor. We have focused on the case of a single model.  Therefore, it is not possible to make a direct comparison with the suggested approaches that increase feature extractors during novel sessions. In fact, the dynamics might be even opposite when multiple models are used (i.e., rank might need to be reduced for each task's model because each model is trained with only one task's dataset.)

Thank you once again for your insightful feedback, and we will resubmit an improved version in the next venue.